# Assessment of Knowledge, Attitudes, and Vaccination Practices Regarding the New RSV Vaccine among Health Professionals in Greece

**DOI:** 10.3390/healthcare12151536

**Published:** 2024-08-02

**Authors:** Dimitrios Papagiannis, Nikolaos Tiganis, Ourania S. Kotsiou, Ioannis C. Lampropoulos, Evangelos C. Fradelos, Foteini Malli, Konstantinos I. Gourgoulianis

**Affiliations:** 1Public Health & Adults Immunization Laboratory, School of Health Sciences, Faculty of Nursing, University of Thessaly, 41110 Larissa, Greece; 2Primary Health Care Postgraduate Program, University of Thessaly, 41500 Larissa, Greece; ntiganis@uth.gr; 3Department of Human Pathophysiology, Faculty of Nursing, University of Thessaly, 41500 Larissa, Greece; okotsiou@uth.gr; 4Respiratory Disorders Laboratory, Faculty of Nursing, University of Thessaly, 41110 Larissa, Greece; i.ch.lampropoulos@gmail.com (I.C.L.); mallifoteini@yahoo.gr (F.M.); 5Laboratory of Clinical Nursing, Department of Nursing, University of Thessaly, 41500 Larissa, Greece; evagelosfradelos@hotmail.com; 6Department of Respiratory Medicine, School of Health Sciences, Faculty of Medicine, University of Thessaly, 41500 Larissa, Greece; kgourg@uth.gr

**Keywords:** respiratory syncytial virus, RSV, vaccines, health professionals

## Abstract

The introduction of a new vaccine into immunization programs represents a significant advancement in the global effort to combat vaccine-preventable diseases. Data from the World Health Organization support that immunization prevents between 2 and 3 million deaths each year across various diseases, underscoring its pivotal role in global health. The present study aims to assess the knowledge, attitudes, and anticipated vaccination practices among health professionals in Central Greece in response to the potential introduction of new Respiratory Syncytial Virus (RSV) vaccination guidelines by the National Vaccines Committee. Among the 450 health professionals solicited for the study, 219 provided responses, yielding a response rate of approximately 55%. A substantial majority (70.3%) accurately identified the vaccine’s current availability, and 62.1% were aware of the current recommendation for RSV vaccination in pregnant women. In response to whether health professionals support the inclusion of an RSV vaccine in the national vaccination program if it becomes commercially available and is recommended by the Greek National Immunization Program, general practitioners showed the most support, with an average score of 4.86 (95% CI, 4.69–5.00), followed by pediatricians at 4.76 (95% CI, 4.63–4.89), pulmonologists at 4.68 (95% CI, 4.36–5.00), and obstetricians at 4.33 (95% CI, 3.95–4.71). Concerning general opinions on vaccinations, a high level of agreement was noted among the majority of health professionals, excluding nurses. Pharmacists recorded the highest agreement, with a perfect score of 5 (CI, 5.00–5.00), followed closely by pediatricians at 4.99 (CI, 4.97–5.00), GPs at 4.95 (CI, 4.85–5.00), pulmonologists at 4.93 (CI, 4.83–5.00), obstetricians at 4.74 (CI, 4.42–5.00), and nurses at 3.80 (CI, 3.06–4.54). A tailored approach to education is needed to ensure that healthcare professionals can communicate more effectively about RSV risks and vaccination benefits, fostering a proactive stance towards disease prevention and patient care. In essence, our study underscores the importance of knowledge in shaping a compassionate and responsive healthcare environment, ready to meet the challenges of RSV head-on.

## 1. Introduction

The introduction of a new vaccine into immunization programs represents a significant advancement in the global effort to combat vaccine-preventable diseases. Vaccination has long been recognized as a cornerstone of public health strategy, successfully reducing the incidence of, and, in case of smallpox, eliminating, life-threatening infectious diseases. This achievement not only saves millions of lives annually world-wide but is also among the most cost-effective public health interventions. According to the World Health Organization, immunization prevents between 2 and 3 million deaths each year across various countries, highlighting its pivotal role in global health [1]. 

The Paramyxoviridae family encompasses a diverse group of viruses, including the Respiratory Syncytial Virus (RSV), measles, parainfluenza 3, Hendra, and Nipah viruses. RSV, along with the human metapneumovirus, is more specifically classified within the sub-family Pneumoviridae. RSV is a leading cause of respiratory tract infections and severe pulmonary disease [2]. It is remarkably prevalent, with nearly all children having been exposed to the virus by the age of two years [3]. In healthy young adults, RSV infections typically manifest as mild upper respiratory tract illnesses [4]. However, its high infectivity rate results in approximately two-thirds of children being infected by their first birthday, and virtually 100% having been exposed by their second year of life. Notably, about 36% of infants under two years of age experience at least two infections, highlighting the need for effective preventive measures [5,6]. 

Despite the availability of some passive prophylaxis options for high-risk groups, there is a pressing demand for more accessible vaccines to reduce both hospitalization rates and the economic impact of RSV [7]. Scientists have been working on RSV vaccines since soon after the virus was discovered in 1956 [8]. Initial attempts in the 1960s, such as the formalin-inactivated RSV vaccine (FI-RSV), resulted in exacerbated illness in vaccinated children following natural infection, halting progress for years [9,10]. However, after decades of research, significant milestones have been achieved. The fast-shifting shape of the F protein on the viral surface has been one of the many obstacles that previous attempts to create an RSV vaccine have had to overcome. In the 1960s, attempts to create a vaccine were unsuccessful. For several decades, the development of an alternative RSV vaccination was impeded by safety concerns. On 3 May 2023, the FDA approved the first RSV vaccine for individuals aged 60 and older, in the fight against this pervasive virus [11]. 

When deciding whether to introduce a vaccine into the national immunization program, decision makers must consider the potential impact of the introduction on both the program and on the overall health system. Existing literature underscores the importance of trust in public health authorities and healthcare professionals in fostering vaccine acceptance and uptake. Studies have shown a positive correlation between trust and the intention to vaccinate [12]. Conversely, there exists a prevalent perception among the public that the risk of disease is low compared to the perceived risk of adverse vaccine effects, creating substantial barriers to vaccination efforts [13,14,15]. Furthermore, the introduction of a vaccine into the national immunization program may also present opportunities to improve the immunization and health systems. For example, the training of health workers for the new vaccine presents opportunities to refresh their skills and knowledge in immunization and other related health services. However, to our knowledge, there are currently few published studies on the validation of tools used to assess vaccine and disease knowledge among healthcare professionals HCPs for RSV in Europe, and there is no study in the literature of healthcare professionals’ intention to recommend RSV vaccination in Greece.

Given this context, our study aims to assess the knowledge, attitudes, and anticipated vaccination practices among health professionals in Central Greece in response to the potential introduction of the new RSV vaccination guidelines by the National Vaccines Committee. This evaluation seeks to inform strategies to enhance vaccine acceptance and implementation, thereby contributing to the broader goal of reducing the burden of RSV and other vaccine-preventable diseases within the community. 

## 2. Materials and Methods

This study was structured as a multicenter, cross-sectional analysis utilizing a population-based questionnaire approach. The target demographic for this investigation included a diverse group of healthcare professionals, namely pediatricians, pulmonologists, obstetricians, general practitioners (GPs), pharmacists, and nurses. These individuals were engaged through self-assessment questionnaires administered via face-to-face interviews from 1 October to 31 December 2023. The recruitment process aimed to encompass a wide array of participants from both public and private health sectors across four prefectures within the region of Thessaly, Greece, leveraging convenience sampling methods to enlist volunteers. The study was conducted in five public hospitals, twenty-one primary healthcare centers, and the private health sector. 

To ensure the study’s robustness and statistical significance, the minimum sample size was meticulously calculated based on the sample size formula for estimating a single proportion under the assumption of a population proportion, with adjustments for the finite population correction factor. This calculation considered the total population size of health professionals in the targeted region (N = 2000), an anticipated frequency of the outcome factor in the population (p) of 90% ± 5%, and set confidence limits at 5% with a 95% confidence level. The formula used was n = [DEFF × Np(1 − p)]/[(d2/Z21 − α/2 × (N − 1) + p × (1 − p)], where DEFF represents the design effect for cluster surveys. On the base of the above assumptions, the minimum required sample size calculated was 130 participants. For the sample in the present study, we used convenience sampling and participants were selected based on availability and willingness to take part in the study (https://www.openepi.com/SampleSize/SSPropor.htm (accessed on 20 September 2023)). Normally, when prevalence is unknown, then it is considered to be a prevalence of 50%. We chose a higher prevalence for the sample because we addressed health professionals such pulmonologists and GPs, where a huge part of their work is to suggest and do routine vaccinations. For this reason, the hypothesized % frequency of outcome factor (vaccinations and knowledge about RSV) in the population (p) was expected to be high, and we preferred to choose a high prevalence of 90%. 

The development of our questionnaire was grounded in an extensive literature re-view on RSV and informed by a preliminary survey of twenty diverse health professionals [16,17,18,19,20,21]. This initial phase aimed to capture a broad understanding of RSV and identify key areas of interest for the new vaccine. Following the pre-survey, the research team critically analyzed the feedback, refining the questionnaire to ensure clarity, relevance, and comprehensiveness. On the questions about the knowledge, the right answer was given a value of 1 and the wrong answer was given a value of 0. The only question excluded from the rating was the first self-report of HCPs about their knowledge of RSV, where the scale was from 1 to 5. The assessment of the reliability of the questionnaire produced a Cronbach’s alpha of 0.719. The final questionnaire comprised three sections designed to evaluate health professionals’ knowledge about RSV, their attitudes towards the new RSV vaccine, and their vaccination practices. This structured approach ensured that the survey was both scientifically robust and aligned with practical considerations, facilitating a deep understanding of healthcare professionals’ perspectives on the new RSV vaccine.

All potential participants were assured that their involvement was strictly voluntary, and all collected data were anonymized, solely utilized for the purposes of this study, and kept confidential. The inclusion criteria mandated that participants were currently practicing health professionals (pediatricians, pulmonologists, obstetricians, general practitioners, pharmacists, nurses), within the specified disciplines and possess the ability to read, comprehend, and complete the questionnaire adequately. Informed written consent was obtained from all participants, affirming their understanding of the study’s nature and their voluntary agreement to take part in the survey. This procedural diligence ensured the ethical integrity of the study while aiming to gather insightful data on the knowledge, attitudes, and vaccination practices concerning the new RSV vaccine among health professionals in Central Greece.

### Statistical Analysis—Questionnaire

Descriptive statistics were calculated as frequency, percentage, mean and standard deviation (SD). Pearson’s chi-square test or Fisher’s exact test were used to compare the two groups. Univariate and multivariate logistic regression were used to assess the relationship between participants’ demographic characteristics, knowledge, attitudes, and practices related to the RSV vaccine and willingness to implement the new RSV vaccine in the future. All reported *p* values were two-tailed. *p*-values < 0.05 were considered as statistically significant. Data were calculated using Pearson’s chi-square test for quantitative variables, and 95% confidence intervals (CIs) were calculated. All analyses were performed using SPSS 29.0 (IBM Corporation, New York, NY, USA).

This study adhered strictly to the ethical guidelines outlined in the Declaration of Helsinki and received approval from the Medical School Ethics Committee of the University of Thessaly, under registration number 499/18-9-2023. Prior to participation, all involved health professionals provided written informed consent, ensuring their voluntary involvement and understanding of the study’s aims and procedures.

## 3. Results

### 3.1. Descriptive Analysis: General Characteristics of the Participants

Among the 450 healthcare professionals solicited for the study, 219 provided responses, yielding a response rate of approximately 55%. The participants had an average age of 48.89 ± 9.2 years, as detailed in Table 1. Gender distribution among respondents was 35% male (77 individuals) and 65% female (142 individuals). Professionally, the participants varied, with pediatricians making up 46% of the sample. Nurses accounted for 16%, pulmonologists 12.8%, obstetricians 12.4%, GPs 10.5%, and pharmacists comprised the remaining 2.3% of the participant pool.

### 3.2. General Knowledge Answers

Table 2 presents the general knowledge answers of the study group. Regarding the newly developed RSV vaccine, approximately two-thirds (64.9%) of healthcare professionals reported having adequate knowledge about the vaccine. A substantial majority (70.3%) accurately identified the vaccine’s current availability. When it comes to vaccine recommendations and target demographics, healthcare professionals demonstrated a high level of accuracy, particularly regarding the advisability of vaccination for adults, pregnant individuals, and adults with comorbidities, with 86.8% providing correct responses. However, only about half (52.1%) correctly answered questions regarding vaccination for infants up to 6 months, children, and adolescents. Concerning the transmission of RSV, about 89% of healthcare professionals correctly understood that RSV cannot be transmitted through food consumption. Additionally, a significant majority (98.2%) correctly identified airborne droplets as a mode of RSV transmission. The peak season for RSV, identified as November through March, was correctly recognized by 67.5% of respondents. While 57% correctly answered that specific treatments for RSV exist, a misconception was observed, with one-third incorrectly suggesting that passive immunity could serve as immune prophylaxis against RSV. Two-thirds (66.7%) were knowledgeable about the timing and dosage for administering passive immunity, yet only half recognized the potential for serious neurological complications from RSV infection. The vast majority (84.5%) correctly believed that antibiotics are ineffective against RSV, and 73.1% knew that maternal antibodies can reduce the risk of RSV infection in infants during their first 4–6 months. Additionally, 62.1% were aware of the current recommendation for RSV vaccination in pregnant women. Lastly, the inquiry regarding the tetanus, diphtheria, and acellular pertussis (TdaP vaccine’s) recommendation for pregnant women as a booster by the Greek national vaccination program served as a checkpoint for overall knowledge about national vaccination guidelines.

### 3.3. General Attitudes Answers 

The attitudes of healthcare professionals (HCPs) surveyed in this study reflect a strong adherence to, and support for, vaccination protocols. An overwhelming majority, 97.3%, adhere strictly to the guidelines set forth by the national immunization program, applying these recommendations to both their personal health management and their clinical practice with patients. Furthermore, 93.1% of HCPs expressed a willingness to recommend the new RSV vaccine once it becomes commercially available and is included in the Greek national immunization program.

When it comes to trust in vaccines, the vast majority 97.2% of healthcare professionals reported a high level of trust in vaccination as an intervention for infection control. This trust extends to the perception of vaccines as a crucial public health tool, with an astonishing 99.1% of respondents recognizing vaccines’ vital role in combating infectious diseases and their significant impact on reducing disease-related mortality and morbidity (Table 2). 

### 3.4. General Practice Questions 

The final two questions in our survey focused on the vaccination practices of healthcare professionals (HCPs), specifically regarding their recommendations to clients and their own vaccination statuses in line with the national immunization program guidelines. When asked if they recommend the TdaP (Tetanus, Diphtheria, acellular Pertussis) vaccine to their clients in accordance with the guidelines set by the Greek national immunization program, a significant majority, 90.9%, reported that they do indeed follow these recommendations and advise their clients accordingly.

Regarding their personal vaccination status, the participants reported high levels of compliance with the national immunization program for adults, specifically, 98.6% of the respondents have been vaccinated against COVID-19, making it the most commonly received vaccine among the surveyed healthcare professionals. This is followed by 71.7% who reported receiving the TdaP vaccine, 65.3% who have been vaccinated against hepatitis B, 61.1% who received the seasonal flu vaccine, and finally, 46.1% who have been vaccinated against pneumococcal disease.

Regarding the question, “How do you rate your knowledge about RSV?” pulmonologists reported the highest average self-assessment score of 4.54 (confidence interval (95% CI, 4.29–4.78), followed by GPs with an average score of 4.29 (95% CI, 3.83–4.74), pediatricians with 4.02 (95% CI, 3.84–4.20), pharmacists with 3.50 (95% CI, 2.58–4.42), obstetricians with 2.81 (95% CI, 2.30–3.33), and finally, nurses with 2.80 (95% CI, 2.31–3.29) (Table 3). In response to whether health professionals support the inclusion of an RSV vaccine in the national vaccination program if it becomes commercially available and is recommended by the Greek National Immunization Program, general practitioners showed the most support, with an average score of 4.86 (95% CI, 4.69–5.00), followed by pediatricians at 4.76 (95% CI, 4.63–4.89), pulmonologists at 4.68 (95% CI, 4.36–5.00), and obstetricians at 4.33 (95% CI, 3.95–4.71).

Concerning general opinions on vaccinations, a high level of agreement was noted among the majority of health professionals, excluding nurses. Pharmacists recorded the highest agreement, with a perfect score of 5 (95% CI, 5.00–5.00), followed closely by pediatricians at 4.99 (95% CI, 4.97–5.00), GPs at 4.95 (95% CI, 4.85–5.00), pulmonologists at 4.93 (95% CI, 4.83–5.00), obstetricians at 4.74 (95% CI, 4.42–5.00), and nurses at 3.80 (95% CI, 3.06–4.54). 

In a univariate analysis focusing on knowledge about and support for the implementation of the new RSV vaccine, significant differences were found favoring men, with *p*-values of 0.024 for the first question and 0.148 for the second, indicating a noteworthy gender disparity in responses (Table 4).

## 4. Discussion

This is the first study for assessment of the knowledge and practices in Greek health professionals regarding the upcoming vaccine. A significant majority of the participants correctly identified the vaccine’s current availability status and disease transmission. Contrasting findings from a similar study among GPs in Italy indicated a lack of satisfactory understanding of RSV and inconsistent risk perceptions, especially concerning RSV infections in elderly populations [20].

It has been acknowledged that healthcare professionals (HCPs) require evidence-based data when discussing the introduction of new vaccines with patients and the general public in order to reassure patients about the shots’ safety and the strength of the regulatory framework. The recent initiation of COVID-19 vaccinations has highlighted that a deficiency in long-term efficacy and safety data were affecting the introduction of new vaccine among both patients and HCPs. Hence, reassurances are often based on extrapolations from existing vaccine data, despite the absence of any recognized mechanism suggesting potential hazards from COVID-19 vaccinations. Prior studies, conducted before the pandemic, revealed a hesitance towards future vaccinations against the coronavirus, with an estimated acceptance rate of 43% [21]. This acceptance was positively correlated with knowledge and attitudes towards preventive measures, suggesting that informed individuals are more likely to perceive vaccinations favorably [22]. We present data on Greece for the healthcare professionals and to intention to vaccinate with the new RSV vaccine. Our data agree with published literature that the better knowledge for the vaccine can be significant associated with future vaccinations [23,24,25,26].

Notably, pulmonologists exhibited the highest knowledge levels about RSV, likely due to the disease’s respiratory involvement, which falls within their specialty’s scope. This expertise is crucial for managing RSV’s varying clinical manifestations, which can range from mild influenza-like symptoms to severe respiratory complications requiring intensive care [27].

The study also explored the influence of the workplace on vaccine-related knowledge, observing that healthcare professionals in the private sector seemed more informed about the RSV vaccine. This variation could reflect the different roles that public and private sectors play in vaccine delivery across countries. In Greece, the private sector, particularly pediatricians and GPs, has been instrumental in administering routine vaccinations to infants and teenagers, in contrast to the mass vaccination campaigns against COVID-19 handled by the public sector. This pattern mirrors the vaccination policies in countries like France, where the private sector plays a significant role in vaccine administration [28,29].

A high number of RSV hospitalizations among adults, notably older adults and persons with specific conditions including transplantation, chronic obstructive pulmonary disease, and heart failure, has been reported. A recent European study, based on the available data, suggests a high burden of RSV in terms of hospital admissions in children aged under 5 years, with most cases occurring among children aged under 1 year [30]. In addition, human respiratory syncytial virus is the leading cause of acute bronchiolitis in infants and young children. A two-period study in hospitalized infants and young children with acute bronchiolitis that was conducted in Greece by Tsergouli et al. recorded high prevalence of RSV; the majority of the cases occurred in January [31].

Vaccination with the new RSV vaccine is one of the cost-effective solutions to prevent the infections and hospitalizations by the virus. The present study highlighted the seasonality of RSV infections in Europe, with a majority of participants accurately identifying the typical onset period relative to the influenza season. This knowledge is crucial for understanding RSV’s impact and preparing for its seasonal peaks.

When asked about their willingness to recommend the new RSV vaccine if included in the Greek national immunization program, a vast majority expressed favorable attitudes. This aligns with findings from other studies and emphasizes the importance of disease burden, vaccine availability, safety, and cost-effectiveness in the decision-making process for new vaccine introductions [32]. Healthcare professionals’ beliefs and willingness to adhere to immunization committee guidelines significantly influence their vaccine uptake and recommendations.

Taking the above into consideration strategies to increase vaccination coverage should thus be customized for the various specialties of HCP, as vaccine confidence levels and factors impacting vaccine acceptance can differ across them [33]. For instance, in spite of the expanding conversations about community pharmacists’ involvement in immunization, particularly with regard to COVID-19 vaccination, there are many research papers examining pharmacists’ views and ideas regarding vaccinations about the acceptability and providing of COVID-19 vaccinations [34,35,36]. While community pharmacists, represented in smaller numbers in this study, show potential as key players in future vaccination strategies, their role in enhancing vaccine acceptance and administration, especially for RSV, warrants further exploration.

The study concludes with a call for vaccination strategies that are tailored to the specific needs and perspectives of different healthcare professionals. The relatively small number of pharmacists participating in the study, yet showing potential as key players in vaccination strategies, underscores the need for inclusive approaches that consider the diverse roles within the healthcare ecosystem. Future strategies should aim to bolster vaccine acceptance and administration capabilities across all healthcare professions, with particular attention to underrepresented groups. In essence, this study not only sheds light on the current state of knowledge and attitudes towards the RSV vaccine among healthcare professionals in Greece but also provides a blueprint for addressing the challenges and opportunities in enhancing vaccine coverage and acceptance. The findings advocate for a multifaceted approach involving education, policy advocacy, and strategic collaborations to ensure the successful introduction and uptake of the RSV vaccine.

The present study demonstrates the relationship between knowledge practices and attitudes regarding the new vaccine against RSV in Greece. However, it is essential to acknowledge certain limitations inherent in our study. Primarily, its cross-sectional design and the non-randomized nature of sampling could potentially impact the universality of the findings. The main limitation is that it is geographically restricted to the Central Greece region and conducted with a relatively small sample size, which leads to limited power and generalizability. Additionally, the focus on Central Greece may limit the applicability of our conclusions across the broader national landscape of healthcare professionals’ perceptions and behaviors.

## 5. Conclusions

RSV is also a threat both to young infants and among immunocompromised and vulnerable individuals. High mortality rates have been observed in those infected with RSV following bone marrow or lung transplantation [37]. According to current research, Greek healthcare professionals had a high degree of RSV knowledge, which was strongly correlated with their favorable attitudes and behaviors about vaccinations against RSV. Results from the present study could be important knowledge related to RSV in patients at increased risk of severe RSV. Findings from the study can be used by public health authorities initiating shared clinical decision-making conversations with other health authorities who are eligible to decide for RSV vaccination to the future. Furthermore, the findings could be useful as well to tailoring RSV disease awareness educational interventions to healthcare providers and patients. Despite these considerations, we firmly believe that the insights garnered provide a reliable snapshot of Greek healthcare professionals’ readiness to embrace and advocate for the new RSV vaccine. Such an investigation not only contributes significantly to our understanding of the factors influencing vaccine uptake among healthcare providers but also highlights the critical need for targeted educational initiatives. These findings advocate for a structured approach to continuing professional development, ensuring that healthcare workers are well-informed and equipped to lead public health responses against RSV and similar infectious threats. Through this lens, the study not only addresses a specific gap in the literature but also sets a precedent for future research and policy making in the realm of vaccine dissemination and public health strategy.

## Figures and Tables

**Table 1 healthcare-12-01536-t001:** Demographics of the participants (N = 219).

Characteristics		
Male, n (%)	77	35%
Female, n (%)	142	65%
Age (years)	48.89 ± 9.2	
Pediatricians, n (%)	101	46%
Pulmonologists, n (%)	28	12.8%
Obstetricians, n (%)	27	12.4%
Nurses, n (%)	35	16%
General Practitioners, n (%)	23	10.5%
Pharmacists, n (%)	5	2.3%
Ph.D., M.Sc., n (%)	108	49%
Undergraduate, n (%)	219	100%
Workplace		
Private, n (%)	128	58%
Public, n (%)	91	425

Abbreviations: Ph.D., Doctor of Philosophy; M.Sc., Master of Science.

**Table 2 healthcare-12-01536-t002:** Given responses on KAP questionnaire (N = 219).

Knowledge Questions		Correct Answer N, %	
1.	How would you assess your level of knowledge regarding Respiratory Syncytial Virus (RSV)?	142	64.9%
2.	Is there a currently available and safe vaccine against RSV?	154	70.3%
3.	Are infants up to 6 months of age targeted by current vaccine recommendations for RSV?	72	32.8%
4.	Do current vaccine recommendations for RSV include children and adolescents?	114	52.1%
5.	Are adults with comorbidities included in the current RSV vaccine recommendations?	190	86.8%
6.	Are pregnant women addressed in the current RSV vaccine recommendations?	100	45.6%
7.	Are individuals older than 60 years targeted by RSV vaccine recommendations?	107	48.8%
8.	Can RSV be transmitted through food consumption?	195	89%
9.	Is RSV transmitted through airborne droplets?	215	98.2%
10.	Do RSV symptoms mirror those of the flu?	188	85.8%
11.	Is there a specific treatment available for RSV?	125	57.1%
12.	In Europe, is the RSV season recognized as spanning from November to March?	148	67.5%
13.	Can currently available passive RSV antibodies be utilized for RSV immune-prophylaxis?	162	73.90%
14.	At what timing and dosage should RSV immune-prophylaxis with available antibodies be administered?	146	66.7%
15.	Can RSV infections lead to severe neurological complications?	116	53.0%
16.	Are antibiotics effective in treating RSV infections?	185	84.5%
17.	Do maternal antibodies decrease the risk of RSV infections in the first 4–6 months of an infant’s life?	160	73.1%
18.	Is the RSV vaccine recommended for pregnant women?	136	62.1%
19.	Is the Diphtheria, Pertussis, Tetanus (DTP) vaccine recommended as a booster for pregnant women within the national immunization program?	210	95.9%
	Attitude Questions	N, %	
20.	Do you adhere strictly to the vaccination recommendations provided by the National Vaccination Program, both personally and for your patients?	213	97.3%
21.	Would you support the inclusion of an RSV vaccine in the national vaccination program if it becomes commercially available and is endorsed by the Greek National Immunization Program (N.I.P.)?	204	93.1%
22.	In general, do you trust vaccines?	213	97.2%
23.	Do you consider vaccines a crucial public health tool against infectious diseases?	217	99.1%
24.	Do you believe that vaccines significantly contribute to the reduction of mortality and morbidity across various diseases?	217	99.1%
	Practice Questions	N, %	
25.	Do you recommend the TdaP vaccine to your clients in line with the national vaccination program’s recommendations?	199	90.9%
26.	Which of the following vaccines, recommended by the national immunization program for adults, have you received?		
	Seasonal Flu	134	61.1%
	Pneumococcus	101	46.1%
	Hepatitis B	143	65.3%
	Diphtheria-Whooping-Tetanus TdaP	157	71.7%
	COVID 19	216	98.6%

Abbreviations: N.I.P., National Immunization Program.

**Table 3 healthcare-12-01536-t003:** Descriptive score criteria for evaluating vaccines by profession.

Question	Profession	Mean Score	St. Error	95% CI	*p*-Value
1. Please rate your level of knowledge regarding Respiratory Syncytial Virus (RSV).	Pulmonologist	4.54	0.120	(4.29–4.78)	<0.001
	Obstetrician	2.81	0.251	(2.30–3.33)	0.011
	GP	4.29	0.220	(3.83–4.74)	<0.001
	Pharmacist	3.50	0.289	(2.58–4.42)	0.024
	Nurse	2.80	0.400	(2.31–3.29)	0.042
	Pediatrician	4.02	0.91	(3.84–4.20)	<0.001
2. Do you support the inclusion of an RSV vaccine in the national vaccination program, should it become commercially available and recommended by the Greek National Immunization Program (N.I.P.)	Pulmonologist	4.68	0.155	(4.36–5.00)	<0.001
	Obstetrician	4.33	0.185	(3.95–4.71)	<0.001
	GP	4.86	0.78	(4.69–5.00)	<0.001
	Pharmacist	N.I *	N.I *	N.I *	-
	Nurse	N.I *	N.I *	N.I *	0.007
	Pediatrician	4.76	0.65	(4.63–4.89)	<0.001
3. Generally, I trust the Vaccinations	Pulmonologist	4.93	0.50	(4.83–5.00)	<0.001
	Obstetrician	4.74	0.156	(4.42–5.00)	<0.001
	GP	4.95	0.48	(4.85–5.00)	<0.001
	Pharmacist	5.00	0.00	(5.00–5.00)	-
	Nurse	3.80	0.327	(3.06–4.54)	0.005
	Pediatrician	4.99	0.10	(4.97–5.00)	<0.001

* No implementation. Abbreviations: GP, General Practitioner.

**Table 4 healthcare-12-01536-t004:** Knowledge and implementation of a new vaccine.

Characteristics	Category	n (%)	1. How Would You Assess Your Level of Knowledge Regarding Respiratory Syncytial Virus (RSV)?	2. Do You Strictly Follow the GUIDELINES and Recommendations Provided by the (N.I.P. **) for Both Yourself and Your Patients?
			*p*-value	
Age (years) *			0.073	0.494
Gender *	Males, n (%)	77 (35.16)	0.024	0.148
	Females, n (%)	142 (64.84)		
Workplace	Private sector	133 (60.73)	0.007	<0.001
	Public sector, n (%)	86 (39.27)		

* Independent *t*-test. ** Greek National Immunization Program.

## Data Availability

The data that support the findings of this study are available on request from the corresponding author.

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
