# Peer review of "Assessment of Knowledge, Attitudes, and Vaccination Practices Regarding the New RSV Vaccine among Health Professionals in Greece"

_healthcare, 2024, doi:10.3390/healthcare12151536_

Round 1

Reviewer 1 Report

Comments and Suggestions for Authors

The paper  is interesting but it has serious error and omissions. I recommend major changes.

1)           Abstract 4.86 (CI, 4.69-5.00,) it should be written if  the confidence interval is 95%. 90% or 99%.  Although the 95% is the most commonly used, there are times when the 99% is used, for example in the estimations with the Monte Carlo Method, in the same way there are epidemiologists such as Rothman who prefer to use the 90% confidence intervals.It is necessary to write the confidence of the confidence intervals (Most probably will be 95% but is should written any way)

2)           INTRODUCTION “….it  has long been recognized as a cornerstone of public health strategy, successfully reducing 44 the incidence of, and in some cases eliminating, life-threatening infectious diseases.” As far as I know there is only one disesease that have been eradicated with vaccination (Small pox) Please correct this sentence.

3)           “ades of re-search, significant milestones have been achieved. Notably, on May 3, 2023, the 67 FDA ap-proved the first RSV vaccine for individuals aged 60 and older, marking a historic 68” Please eliminate the hyphens.

4)           The authors say “The results of multivariate logistic 123 regression analyses were presented in forest plots with the comparative odds ratio (OR) 124 and 95% confidence intervals (CIs).” But there are no OR, neither logistic regression.

5)           The same happen with ANOVA. There is not ANOVA in the text.

6)           Please explain better the process of sample selection, simple random, stratified, cluster ....

7)           It is not clear on the one hand, some sample size calculations are presented indicating that there are 2000 health professionals and that 130 are needed, but later on, it is not clear how because they say that  of the 450 health professionals solicited for the study, 219 provided responses.  A graph would be useful. 

8) Explain the difference between DTP and TDaP, so that a non expert can understand the differences between themz

Author Response

Reviewer 1

The paper is interesting but it has serious error and omissions. I recommend major changes.

1)           Abstract 4.86 (CI, 4.69-5.00,) it should be written if the confidence interval is 95%. 90% or 99%.  Although the 95% is the most commonly used, there are times when the 99% is used, for example in the estimations with the Monte Carlo Method, in the same way there are epidemiologists such as Rothman who prefer to use the 90% confidence intervals. It is necessary to write the confidence of the confidence intervals (Most probably will be 95% but is should written anyway)

Authors response: We would like to thank the reviewer for the comment we added 95% CI in the abstract section.

2)           INTRODUCTION “….it  has long been recognized as a cornerstone of public health strategy, successfully reducing 44 the incidence of, and in some cases eliminating, life-threatening infectious diseases.” As far as I know there is only one disease that have been eradicated with vaccination (Small pox) Please correct this sentence.

Authors response: We would like to thank the reviewer for the comment. We modified the text accordingly.

3)           “ades of research, significant milestones have been achieved. Notably, on May 3, 2023, the FDA approved the first RSV vaccine for individuals aged 60 and older, marking a historic” Please eliminate the hyphens.

 Authors response: Done

4)           The authors say “The results of multivariate logistic 123 regression analyses were presented in forest plots with the comparative odds ratio (OR) 124 and 95% confidence intervals (CIs).” But there are no OR, neither logistic regression.

 Authors response: thanks the reviewer for the constructive comment, we apologized for the error. We have done the analysis before the submission and our data didn’t report statistically significant results. We analyzed the OR for the professions, please see in the follow table. Profession (1): The odds ratio (Exp(B)) for Profession (1) is 1.533, to the reference category. However, this result is not statistically significant (p = 0.464), and the confidence interval (0.488 to 4.813) includes 1, indicating that the result may not be reliable and we preferred to excluded of the results. However, we deleted from the text the OR and modified the text accordingly.  

5)           The same happen with ANOVA. There is not ANOVA in the text.

Authors response: done

6)           Please explain better the process of sample selection, simple random, stratified, cluster ....

Authors response: We would like to thank the reviewer for the comment. We added the phrase “About the sample in the present study we used Convenience sampling and participants are selected based on availability and willingness to take part in the study”.

7)           It is not clear on the one hand, some sample size calculations are presented indicating that there are 2000 health professionals and that 130 are needed, but later on, it is not clear how because they say that of the 450 health professionals solicited for the study, 219 provided responses.  A graph would be useful.

Authors response: thank the reviewer for the comment. We use the open epi system to calculate the minimum size sample of the total pool of 2000 health professionals of Central Greece according to Greek national immunizations program could suggest vaccinations. We addressed to 450 health professionals and they responded 219. Explained by the graphical abstract were included to initial submission. Please find attached the graphical abstract and the link      of open      epi and how calculated the sample.                       (https://www.openepi.com/SampleSize/SSPropor.htm).

8) Explain the difference between DTP and TdaP, so that a non-expert can understand the differences between themz

Authors response: Thank the reviewer for the comment. We explain the abbreviations of Tdap in lines 175-176, line 196, we added the phrase “the tetanus, diphtheria, and acellular pertussis (TdaP vaccine's)”

Reviewer 2 Report

Comments and Suggestions for Authors

The authors tried to assess the level of knowledge, attitudes, and practices of health professionals  in Greece regarding RSV vaccine. The subject is interesting and would provide interesting information regarding the phenomenon of vaccine acceptance/hesitancy.

After reviewing, I found however some gaps that has affected the quality of the manuscript.

First : the manuscript lacks a certain rigor regarding the way of writing showing multiple spelling errors (i.e: " re-search": line 67, "ap-proved": line 68, line 96….), punctuation… and incomplete sentences.

2. the description of the results is not in accordance with the content of the tables (i.e: line 162: 52.1% and line 163: 89%...). In addition,  the order of the description of the results is not coherent: he authors begin with the knowledge about the vaccine, the provide information regarding the disease and after that they return to the vaccine knowledge. Revise (the paragraphs and the content of table 1).

3. the authors did not provide how did they calculate the mean of knowledge. In addition, one and not understand when the mean is self-reported and when t was calculated by the authors.

4. The analysis done in the last paragraph (table 4) could be done with the calculated level of knowledge regarding both the disease and the vaccine (you can calculate the level of knowledge regarding the disease and the vaccine from table 1). They could also completed with other statistical analysis (regression..).

5. When we delete a paragraph(from the discussion) we should revise the other paragraphs accordingly (we cannot begin a paragraph with " regarding their knowledge..). also, the content of the following paragraph (line 272-274) is not in accordance with your results (You are discussing the level of knowledge about the vaccine or about the disease?). In line 281, it may be RSV vaccine not coronavirus vaccine,. In addition n the paragraph : "Our finding for Greece.." is not in accordance with your results. Have you studied the relation between the knowledge and the attitude? You should revise and summaries (avoid long self-paragraphs ) your discussion extensively.

I have also other remarks:

In the introduction, you should provide what was done before regarding this chapter (RSV vaccine knowledge and attitude) in Greece and in the world to define the hypothesis of the work.

Line 98: why did you choose the proportion of 90%? Provide the reference of the formula or the site used to calculate he sample size.

Lines 127-142: the content of the questionnaire should be provided before the statistical analysis.

Lines 123-126: where are the results of OR in the text? Have you really done these analyses?

You should provide how did you calculate the level of knowledge.

In table 1 delete n (%) from the variables since it has been provided in the first line.

You should avoid to indicate each table  multiple times I the results

What did Figure one provide in particular? Why exactly for the variable "sex"? delete.

Lines 261-265: delete this paragraph (you are in the results no in the discussion).

Table 3: not implementation should be "no implementation".

Line 360: what do you mean by "impressive"? what is your criteria to say so?

Your conclusion should be completely revised by providing the most important results of the study.

In conclusion, the manuscript lacks a certain mastery and rigor and should be extensively revised

Comments on the Quality of English Language

Extensive Editing required

Author Response

 Reviewer 2

The authors tried to assess the level of knowledge, attitudes, and practices of health professionals in Greece regarding RSV vaccine. The subject is interesting and would provide interesting information regarding the phenomenon of vaccine acceptance/hesitancy.

After reviewing, I found however some gaps that has affected the quality of the manuscript.

First : the manuscript lacks a certain rigor regarding the way of writing showing multiple spelling errors (i.e: " re-search": line 67, "ap-proved": line 68, line 96….), punctuation… and incomplete sentences.

Authors response: Authors response: Thank the reviewer for the comment. We modified the text accordingly.

  1. the description of the results is not in accordance with the content of the tables (i.e: line 162: 52.1% and line 163: 89%...). In addition, the order of the description of the results is not coherent: the authors begin with the knowledge about the vaccine, the provide information regarding the disease and after that they return to the vaccine knowledge. Revise (the paragraphs and the content of table 1).

Authors response: Thank the reviewer for the constructive comment. We apologize for the type error. We modified the text and table accordingly.

  1. the authors did not provide how did they calculate the mean of knowledge. In addition, one and not understand when the mean is self-reported and when t was calculated by the authors.

Authors response: Thank the reviewer for the comment. According to questions true or false answer evaluated for each health professional supported by international data. In the self-report question “Please rate your level of knowledge regarding Respiratory Syncytial Virus RSV” the scale was 1 to 5 with the 1as low level of knowledge and 5 as high level of knowledge. We calculate the mean of knowledge by dividing the sum of all the numbers within the data set by the number of data points for each specialty.

  1. The analysis done in the last paragraph (table 4) could be done with the calculated level of knowledge regarding both the disease and the vaccine (you can calculate the level of knowledge regarding the disease and the vaccine from table 1). They could also completed with other statistical analysis (regression.).

Authors response: Thank the reviewer for the comment. We have done the suggested analysis before the submission and our data didn’t report statistically significant results. The regression analysis didn’t report statistically significant results also. And for these reasons we decide to exclude this analysis.  Please find attached the analysis of q2 and q11.

Q2.” Is there an available and safe vaccine against syncytial virus-RSV? and Q11.There is specific treatment available for Respiratory syncytial virus (RSV).

  1. When we delete a paragraph (from the discussion) we should revise the other paragraphs accordingly (we cannot begin a paragraph with " regarding their knowledge..). also, the content of the following paragraph (line 272-274) is not in accordance with your results (You are discussing the level of knowledge about the vaccine or about the disease?). In line 281, it may be RSV vaccine not coronavirus vaccine, In addition n the paragraph: "Our finding for Greece.." is not in accordance with your results. Have you studied the relation between the knowledge and the attitude? You should revise and summaries (avoid long self-paragraphs ) your discussion extensively.

Authors response: Thank the reviewer for the comment. We modified the text and deleted the phrase “Regarding their knowledge on the new RSV vaccine”

And reform the 2nd phrase to “findings of the present study highlight that better knowledge.”

I have also other remarks:

In the introduction, you should provide what was done before regarding this chapter (RSV vaccine knowledge and attitude) in Greece and in the world to define the hypothesis of the work.

Authors response: Thank the reviewer for the comment. We modified the text and added the phrases

  1. When deciding whether to introduce a vaccine into the national immunization programme, decision-makers must consider the potential impact of the introduction on both the programme and on the overall health system.
  2. Furthermore, the introduction of a vaccine into the national immunization program may also present opportunities to improve the immunization and health systems. For example, the training of health workers for the new vaccine presents opportunities to refresh their skills and knowledge in immunization and other related health services. To our knowledge, there is no exploration in the literature of health care professional’s intention to recommend RSV vaccination to at risk patients in Greece.

Line 98: why did you choose the proportion of 90%? Provide the reference of the formula or the site used to calculate he sample size.

Authors response: thanks, the reviewer for the comment. We choose 90% proportion for CI 95% because addressed to health professions whose targeted on vaccinations. According to Greek national immunizations program these 5 specialties of HP could suggest the vaccinations.  For these reason chosen high proportion. We added the link of size calculation in to the text. (https://www.openepi.com/SampleSize/SSPropor.htm )

Lines 127-142: the content of the questionnaire should be provided before the statistical analysis.

Authors response: Thank the reviewer for the comment. We transfer the paragraph before the statistical analysis.

Lines 123-126: where are the results of OR in the text? Have you really done these analyses?

Authors response: We agree with the reviewer for this statement and apologize. We analyzed the OR for the professions, please see in the follow table.

a

Profession (1): The odds ratio (Exp(B)) for Profession (1) is 1.533, to the reference category. However, this result is not statistically significant (p = 0.464), and the confidence interval (0.488 to 4.813) includes 1, indicating that the result may not be reliable and we preferred to excluded of the results. However, we deleted from the text the OR.

You should provide how did you calculate the level of knowledge.

Authors response: According to answer (false or true) we evaluated the answers based to international literature. For example, about the question “Do current vaccine recommendations for RSV include children and adolescents?” the right answer is No. According to CDC and FDA CDC recommends an RSV vaccine for everyone ages 75 and older and adults ages 60-74 at increased risk of severe RSV. Adults 60-74 who are at increased risk include those with chronic heart or lung disease, certain other chronic medical conditions, and those who are residents of nursing homes or other long-term care facilities. Vaccination for pregnant people 1 dose of maternal RSV vaccine during weeks 32 through 36 of pregnancy, administered September through January. Source CDC available at:                           https://www.cdc.gov/vaccines/vpd/rsv/index.html

In table 1 delete n (%) from the variables since it has been provided in the first line.

Authors response: Done

You should avoid to indicate each table multiple times I the results

Authors response: Thank the reviewer for the comment. We modified the text accordingly

What did Figure one provide in particular? Why exactly for the variable "sex"? delete.

Authors response: Done

Lines 261-265: delete this paragraph (you are in the results no in the discussion).

Authors response: Done

Table 3: not implementation should be "no implementation".

Authors response: Done

Line 360: what do you mean by "impressive"? what is your criteria to say so?

Authors response: Thank the reviewer for the comment. We deleted.

Your conclusion should be completely revised by providing the most important results of the study.

Authors response: Thank the reviewer for the comment. We reformed all section of conclusions accordingly with a new reference [37] and we added the paragraph “The risk of severe disease in adults is increased by the presence of underlying chronic pulmonary disease, circulatory conditions and functional disability, and is associated with higher viral loads. RSV is also a threat both to young infants and among immunocompromised and vulnerable individuals. High mortality rates have been ob-served in those infected with RSV following bone marrow or lung transplantation [37]. According to current research, Greek healthcare professionals had a high degree of RSV knowledge, which was strongly correlated with their favorable attitudes and behaviors about vaccinations against RSV. Results from the present study could be important knowledge related to RSV in patients at increased risk of severe RSV. Findings from the study can be used by public health authorities initiating shared clinical decision-making conversations with other health authorities who are eligible to decide for RSV vaccination to the future. Furthermore, findings could be useful as well as to tailor RSV disease awareness educational interventions to healthcare providers and patients”.

In conclusion, the manuscript lacks a certain mastery and rigor and should be extensively revised.

Authors response: Thank the reviewer for the comment. We tried to modified the text.

Round 2

Reviewer 2 Report

Comments and Suggestions for Authors

The authors tried to revise their manuscript. However they failed to answer to most of my comments (their responses are not in accordance with the content of the manuscript). The manuscript is still lacking a certain rigor.

The same errors are apparent (line 14, exposed in line 55, research in line 120, punctuation in line 18 (variables), SPSS 29.0 in line 149……, the font character used in table 1, line 169-171, )

You should provide more results in the abstract

The recommendations should be provided after the conclusion.

Introduction:

You have not provided what was done before in your country and in the world regarding RSV vaccine attitudes and knowledge

Methods:

You should explain why did you choose a proportion of 90%.

Results:

The same errors are observed: you should provide the knowledge about the disease and then the vaccine (logically)

The main concern is why did you ask all these questions if they are not used to calculate the level of knowledge (since you asked them to rate their level of knowledge). Even the question of self assessment is interesting, the most interesting is to calculated the level of knowledge from the asked question that could be used in the further analyses.

Delete the sentence of line 240-241 or complete it and include its results in the table.

The paragraph of line 223-237 is repeated in lines 242-256

Revise and summarize the paragraphs of lines 264-281 (do not discuss the results).

The discussion is still very superficial and the comparisons are misleading (example: lines 286-20: A significant majority…elderly populations". Here you are comparing your results (about the vaccine) with the results of another study (about the knowledge about the disease).

Lines 296-297: the same error persists (RSV not COVID-19).

Line 301-302: "The current study highlight that better knowledge about a vaccine significantly influences the intention to vaccinate" . the same error persists: Have you really study the relation between the level of knowledge and attitude?

The discussion should be completely revised

Conclusion:

The first paragraph (lines 378-381) should be deleted.

At last, the manuscript should undergo an exstensive language editing

Comments on the Quality of English Language

Extensive editing required

Author Response

Reviewer

 The authors tried to revise their manuscript. However, they failed to answer to most of my comments (their responses are not in accordance with the content of the manuscript). The manuscript is still lacking a certain rigor.

The same errors are apparent (line 14, exposed in line 55, research in line 120, punctuation in line 18 (variables), SPSS 29.0 in line 149……, the font character used in table 1, line 169-171, )

Authors response: Thank the reviewer, we modified the text.

You should provide more results in the abstract

Authors response: thank the reviewer for the comment.  We added the part” Concerning general opinions on vaccinations, a high level of agreement was noted among the majority of health professionals, excluding nurses. Pharmacists recorded the highest agreement with a perfect score of 5 (CI, 5.00-5.00), followed closely by pediatricians at 4.99 (CI, 4.97-5.00), GPs at 4.95 (CI, 4.85-5.00), pulmonologists at 4.93 (CI, 4.83-5.00), obstetricians at 4.74 (CI, 4.42-5.00), and nurses at 3.80 (CI, 3.06-4.54) “

The recommendations should be provided after the conclusion.

Authors response: Thank the reviewer for the comment. We modified the text accordingly.

Introduction: You have not provided what was done before in your country and in the world regarding RSV vaccine attitudes and knowledge

Authors response: Thank the reviewer for the comment, we notice that twice. Lines 91-94 introduction section “However, to our knowledge, there is currently published a few studies on the validation of tools used to assess vaccine and disease knowledge among HCPs for RSV in Europe and there is no study in the literature of health care professional’s intention to recommend RSV vaccination in Greece” and lines 285-286 “that this is the first study in Greece to evaluate the knowledge of new RSV Vaccine”.

Methods: You should explain why did you choose a proportion of 90%.

Authors response: We added the part “Normally when prevalence is unknow then it is considered to put prevalence of 50%.  We choose higher prevalence for the sample because we addressed to health professions such a Pulmonologists, GPs, where a huge part of their work is to suggest and do routine vaccinations. For this reason, the Hypothesized % frequency of outcome factor (vaccinations and knowledge RSV) in the population (p) were expected high and we prefer to choosing high prevalence 90%”.  Lines 121-126 in methodology.

Results: The same errors are observed: you should provide the knowledge about the disease and then the vaccine (logically)

Authors response: Thank the reviewer for the comment. We added the part “on the questions about the knowledge the right answer calculated with 1 and the wrong answer with 0”. The only question was excluded from this rating was the first self-report of HCPs question about the knowledge of RSV were the scale was 1 to 5”. Furthermore, we provide the knowledge about the vaccine and disease in line 184 according to self-report question the HP answered that “64.9%) of healthcare professionals reported having adequate knowledge about the vaccine” and Lines 191-201 “However, only about half (52.1%) correctly answered questions regarding vaccination for infants up to 6 months, children, and adolescents. Concerning the transmission of RSV, about 89% of healthcare professionals correctly understood that RSV cannot be trans-mitted through food consumption. Additionally, a significant majority (98.2%) correctly identified airborne droplets as a mode of RSV transmission. The peak season for RSV, identified as November through March, was correctly recognized by 67.5% of respondents. While 57% correctly answered that specific treatments for RSV exist, a misconception was observed, with one-third incorrectly suggesting that passive immunity could serve as immune prophylaxis against RSV. Two-thirds (66.7%) were knowledgeable about the timing and dosage for administering passive immunity, yet only half recognized the potential for serious neurological complications from RSV infection”

The main concern is why did you ask all these questions if they are not used to calculate the level of knowledge (since you asked them to rate their level of knowledge). Even the question of self-assessment is interesting, the most interesting is to calculated the level of knowledge from the asked question that could be used in the further analyses.

Authors response: Thank the reviewer for the comment. On the questions about the knowledge the right answer calculated with 1 and the wrong answer with 0. The only question was excluded from this rating was the first self-report question about the knowledge of RSV were the scale was 1 to 5” and we added on the Materials and Methods section.

Delete the sentence of line 240-241 or complete it and include its results in the table.

Authors response: done

The paragraph of line 223-237 is repeated in lines 242-256

Authors response: thank the reviewer for the observation we deleted.

Revise and summarize the paragraphs of lines 264-281 (do not discuss the results).

Authors response: done

The discussion is still very superficial and the comparisons are misleading (example: lines 286-20: A significant majority…elderly populations". Here you are comparing your results (about the vaccine) with the results of another study (about the knowledge about the disease).

Lines 296-297: the same error persists (RSV not COVID-19).

Authors response: we added the phrase “and disease transmission”

Line 301-302: "The current study highlight that better knowledge about a vaccine significantly influences the intention to vaccinate". the same error persists: Have you really study the relation between the level of knowledge and attitude?

Authors response:  we deleted the phrase.

The discussion should be completely revised

Authors response:  We added 2 new parts accordingly: “It has been acknowledged that health care professionals (HCPs) require evidence based data when discussing the introduction of new vaccines with patients and the general public in order to reassure patients about the shots' safety and the strength of the regulatory framework” lines 323-326 and “were affect the introduction of new vaccine both patients and HCPs” Line 328. However, we mentioned the aim of the study lines 258-59, we summarize the main findings lines 276-77, 281-287, 304-06 and addressing limitations—their potential impact on the result lines 334-341.

Conclusion: The first paragraph (lines 378-381) should be deleted.

Authors response: Done

At last, the manuscript should undergo an exstensive language editing

Authors response: Professional English language editing was performed before our first upload. However, it would be helpful if you could advise us which parts of the manuscript need further editing so as proceed with a second editing by another professional English language editor.
